# Towards Scalable Persistence-Based Topological Optimization

**Abderrahim Bendahi** *
École Polytechnique
Paris, France
abderrahim.bendahi@polytechnique.edu

**Alexandre Duplessis** *
ENS Ulm, PSL
Paris, France
alexandre.duplessis@ens.psl.eu

**Arnaud Fickinger**
UC Berkeley
Berkeley, CA, USA
arnaud.fickinger@berkeley.edu

## Abstract

Persistence-based topological optimization deforms a point cloud $X \subset \mathbb{R}^d$ by minimizing objectives of the form $L(X) = \ell(\mathrm{Dgm}(X))$, where $\mathrm{Dgm}(X)$ is a persistence diagram. In practice, optimization is limited by two coupled issues: persistent homology is typically computed on subsamples, and the resulting topological gradients are highly sparse, with only a few *anchor* points receiving nonzero updates. Motivated by diffeomorphic interpolation, which extends sparse gradients to smooth ambient vector fields via Reproducing Kernel Hilbert Space (RKHS) interpolation, we propose a more scalable pipeline that improves both subsampling and gradient extension. We introduce subsampling via random slicing, a lightweight scheme that promotes iteration-wise geometric coverage and mitigates density bias. We further replace the costly kernel solve with a fast *Nadaraya-Watson* (NW) Gaussian convolution, producing a globally defined smooth update field at a fraction of the computational cost, while being more suited for topological optimization tasks. We provide theoretical guarantees for NW smoothing, including anchor approximation bounds and global Lipschitz estimates. Experiments in 2D and 3D show that combining random slicing with NW smoothing yields consistent speedups and improved objective values over other baselines on common persistence losses.

## 1 Introduction

Persistent homology (PH) provides a multiscale summary of the topology of a dataset by tracking the birth and death of homological features along a filtration, producing a persistence diagram $\mathrm{Dgm}(X)$, and enjoys well-quantified stability properties in bottleneck and Wasserstein distances as established in Chazal et al. (2014); Skraba & Turner (2020). This representation has proved expressive and robust enough to serve as a building block for learning pipelines, where topological summaries enter either as features or as loss terms. A particularly direct use is *persistence-based topological optimization*, where one updates the data itself by minimizing

$$\min_{X=\{x_i\}_{i=1}^n \subset \mathbb{R}^d} L(X) := \ell(\mathrm{Dgm}(X)), \tag{1}$$

with losses $\ell$ encoding tasks such as simplification (remove features), augmentation (create features), or registration to a target diagram, in the spirit of the topological loss design used in Liu et al. (2022).

Despite favorable stability and almost-everywhere differentiability properties, the practical performance of equation 1 is often limited by *gradient sparsity*: at a given iterate, only a small set of points participating in critical birth and death simplices receive nonzero gradients, while most points have zero update, a phenomenon documented for point-cloud optimization in Carrière et al. (2024).

---

*Equal contribution.

A recent remedy is *diffeomorphic interpolation*, introduced in Carrière et al. (2024), which constructs a smooth ambient vector field matching the sparse topological gradient on a set of anchor points and then transports the full point set along the induced flow. This approach turns sparse gradients into global deformations with controlled Lipschitz constants and combines well with subsampling, since the diffeomorphism learned on a subsample can be applied to the entire point cloud. However, in its RKHS formulation, diffeomorphic interpolation requires solving a kernel linear system at each iteration, which can dominate runtime in large-scale settings, and when combined with uniform subsampling—a common choice for scalability—it inherits the high-variance behavior and density bias of the anchors,

**Our Contributions.**

- We propose random slicing to improve subsampling geometric coverage.
- We replace the kernel solve by NW Gaussian smoothing to smooth the gradient.
- We provide theoretical guarantees and validate speed and quality gains on 2D and 3D tasks.

## 2 RELATED WORKS

**Persistence diagrams: algorithms and stability.** The practical use of persistence diagrams relies on their robustness: small perturbations of the input induce small changes in the diagram, formalized in Cohen-Steiner et al. (2007). On the algorithmic side, Zomorodian & Carlsson (2005); Edelsbrunner et al. (2002) established the algebraic and computational foundations that underlie most modern PH pipelines.

**Scaling persistent homology.** Exact Vietoris–Rips PH becomes prohibitive at scale; efficient implementations such as Bauer (2021) reduce memory and time constants dramatically, while approximation schemes such as Sheehy (2013) justify sparsifying the filtration itself. These developments motivate subsampling and approximation inside optimization loops, but they also highlight that *how* we subsample can materially affect the optimization trajectory.

**Topological losses in machine learning (ML).** Topological losses have been deployed broadly in ML, for instance as regularizers that enforce desired topological structure in predictions (Chen et al., 2019; Clough et al., 2020), as constraints for geometry-aware representation learning such as *Topological Autoencoders* Moor et al. (2020), and as objectives in shape and point-cloud optimization (Carrière et al., 2021). Beyond regularization, persistence-based losses have been used for segmentation in medical imaging (Clough et al., 2020) and for uncertainty and shift detection via activation-graph topology (Lacombe et al., 2021). In contrast, we focus on directly optimizing point clouds under persistence-based losses, where gradients are typically sparse.

**From sparse topological gradients to smooth ambient updates.** A recent approach to address sparsity is *Diffeomorphic interpolation* (Carrière et al., 2024), which constructs a minimum-norm RKHS vector field matching the sparse anchor gradients and updates points through the induced diffeomorphic flow. Our work follows the same high-level principle (extend sparse gradients to ambient smooth vector fields) but targets scalability: we replace exact kernel interpolation (requiring repeated kernel linear solves) with a fast normalized Gaussian convolution, i.e. a Nadaraya-Watson type estimator originating in *On Estimating Regression* (Nadaraya, 1964) and *Smooth Regression Analysis* (Watson, 1964).

## 3 BACKGROUND

### 3.1 POINT CLOUDS, FILTRATIONS, AND PERSISTENCE DIAGRAMS

Let $X = \{x_i\}_{i=1}^n \subset \mathbb{R}^d$ be a point cloud. We consider the Vietoris–Rips filtration $\{K_t(X)\}_{t \geq 0}$ defined by

$$K_t(X) := \left\{ \sigma \subseteq [n] : \|x_p - x_q\|_2 \leq t, \quad \forall p, q \in \sigma \right\}.$$

For a fixed homological dimension $k \geq 0$, persistent homology yields a multiset of pairs

$$\mathrm{Dgm}_k(X) = \{(b_\alpha, d_\alpha)\}_{\alpha \in \mathcal{I}_k}, \qquad 0 \leq b_\alpha \leq d_\alpha \leq +\infty,$$

where $(b_\alpha, d_\alpha)$ records the birth and death scales of a $k$-dimensional topological feature. We denote the persistence (distance to the diagonal) by $\mathrm{pers}_\alpha := d_\alpha - b_\alpha$.

### 3.2 PERSISTENCE-BASED OBJECTIVES

We study objectives that factor through persistence diagrams:

$$\min_{X \subset \mathbb{R}^d} L(X) \quad \text{where} \quad L(X) := \ell(\mathrm{Dgm}_k(X)), \tag{2}$$

for some loss $\ell$ defined on persistence diagrams. Common examples include:

$$\textit{(simplification)} \quad \ell(\mathrm{Dgm}) = \sum_{\alpha \in \tilde{\mathcal{I}}} \mathrm{pers}_\alpha^2, \qquad \textit{(augmentation)} \quad \ell(\mathrm{Dgm}) = -\sum_{\alpha \in \tilde{\mathcal{I}}} \mathrm{pers}_\alpha^2,$$

and *diagram registration* losses $\ell(\mathrm{Dgm}) = W_p(\mathrm{Dgm}, \mathrm{Dgm}_{\mathrm{target}})$. In practice, $\tilde{\mathcal{I}}$ can select the $k$ most persistent points or a task-specific subset.

### 3.3 SPARSITY OF TOPOLOGICAL GRADIENTS

The map $X \mapsto \mathrm{Dgm}_k(X)$ is stable (Cohen-Steiner et al., 2007; 2010; Skraba & Turner, 2020), and for standard choices of $\ell$, the objective $L$ is Lipschitz and thus differentiable almost everywhere (*Rademacher theorem*, Morgan (2016)). At points of differentiability, the chain rule takes the schematic form

$$\frac{\partial L}{\partial x_i}(X) = \sum_{\alpha : x_i \rightsquigarrow (b_\alpha, d_\alpha)} \left( \frac{\partial \ell}{\partial b_\alpha} \frac{\partial b_\alpha}{\partial x_i} + \frac{\partial \ell}{\partial d_\alpha} \frac{\partial d_\alpha}{\partial x_i} \right), \tag{3}$$

where the sum runs over diagram points whose birth and death critical simplices involve $x_i$. For most indices $i \in [n]$, no participation occurs in equation 3, hence $\nabla L(X)_i = 0$. We call *anchors* the indices with nonzero gradient:

$$I(X) := \{i \in [n] : \nabla L(X)_i \neq 0\}.$$

Gradient sparsity slows optimization and amplifies variance when PH is computed on subsamples.

### 3.4 DIFFEOMORPHIC INTERPOLATION AS AMBIENT SMOOTHING

A principled way to mitigate sparsity is to extend anchor gradients to a smooth ambient vector field $v : \mathbb{R}^d \to \mathbb{R}^d$ and update all points using this field. In particular, diffeomorphic interpolation constructs the minimum-norm RKHS vector field satisfying the interpolation constraints

$$v(x_i) = g_i \quad \text{for all } i \in I, \qquad \text{where} \quad g_i := \nabla L(X)_i, \tag{4}$$

by solving

$$v^\star \in \arg\min_{v \in \mathcal{H}} \|v\|_{\mathcal{H}} \qquad \text{under the constraint 4.} \tag{5}$$

With a Gaussian kernel $K(x, y) = \exp(-\|x - y\|^2 / (2\sigma^2)) I_d$, the *representer theorem* yields

$$v^\star(x) = \sum_{i \in I} K(x, x_i) a_i, \qquad \text{with coefficients } a = (a_i)_{i \in I} \text{ solving } \mathbb{K}a = g,$$

where $\mathbb{K} = (K(x_i, x_j))_{i,j \in I}$. This produces a smooth field that matches anchors exactly, but requires solving a kernel system (typically cubic in $|I|$) at every iteration.

## 4    Towards Scalable Diffeomorphic Topological Optimization

### 4.1    Subsampling via Random Slicing

**Motivation and First Guarantees.**    Computing persistent homology on the full cloud is often prohibitive, so we rely on subsampling. However, in an optimization loop, the subsampling distribution directly affects (i) the variance of the induced anchor gradients and (ii) the geometric regions that can influence topological events. Uniform sampling (used in Carrière et al. (2024)) is cheap but tends to oversample dense regions and miss sparse boundary areas (Figure 1) where long-range simplices often form. We propose a simple structured sampler that enforces global coverage *without* requiring clustering, farthest-point sampling, or additional geometry preprocessing.

Our design is inspired by *sliced* methods in optimal transport (Rabin et al., 2012; Bonneel et al., 2015), which approximate high-dimensional geometry by repeatedly probing random one-dimensional projections where computations reduce to sorting in $\mathbb{R}$. We adopt the same principle for subsampling: each iteration uses one random *slice* to construct a globally spread subset.

More formally, let $X = \{x_i\}_{i=1}^n \subset \mathbb{R}^d$ and fix a target subsample size $s \in \{2, \ldots, n\}$. Draw a random direction $u \sim \mathcal{N}(0, I_d)$ and normalize $u \leftarrow u/\|u\|_2$. Compute the 1D projections

$$t_i := \langle x_i, u \rangle, \qquad i \in [n],$$

and let $\pi$ be a permutation that sorts these values increasingly: $t_{\pi(0)} \leq t_{\pi(1)} \leq \cdots \leq t_{\pi(n-1)}$. We then select indices at evenly spaced ranks:

$$S(u) := \Big\{ \pi(r_k) \ : \ r_k = \lfloor k(n-1)/(s-1) \rfloor, \ \ k = 0, \ldots, s-1 \Big\}. \tag{6}$$

The resulting subset $S(u)$ is used for PH computation at that iteration.

Intuitively, in one dimension, selecting evenly spaced ranks enforces a strong notion of coverage. Thus, for a fixed slice, the sampler avoids the density bias of uniform sampling: extremely dense regions do not *steal* all sample budget, since each interval of the sorted list receives roughly the same number of points.

This notion of coverage can be formalized by the next lemma.

**Lemma 4.1** (Maximal rank gap). *Let $r_k = \lfloor k(n-1)/(s-1) \rfloor$ be the selected ranks.*

$$\max_{k \in \{0, \ldots, s-2\}} (r_{k+1} - r_k) \leq \left\lceil \frac{n-1}{s-1} \right\rceil.$$

Lemma 4.1 implies that, in the sorted 1D order induced by $u$, no large contiguous block of ranks is skipped: every rank is within $\mathcal{O}(n/s)$ positions of a selected one. This does not guarantee Euclidean covering in $\mathbb{R}^d$, but it provides an efficient, iteration-wise notion of global coverage aligned with the slice geometry that empirically stabilizes PH-based optimization. The proof is deferred to Appendix B. Computationally, the sampler requires one projection pass ($\mathcal{O}(nd)$) and a sort ($\mathcal{O}(n \log n)$), typically negligible compared to repeated PH computations.

**Geometric Intuition.**    Figure 1 illustrates how projection-stratified (random slicing) subsampling mitigates the density bias of uniform sampling. The cloud is a mixture of two separated clusters with highly imbalanced masses (85% vs. 15%). Uniform subsampling concentrates almost entirely on the dominant cluster, so the minority cluster may be represented by very few points for small budgets. Random slicing instead selects evenly spaced ranks in a random 1D projection; the separation between clusters induces a gap in projected values, so rank stratification tends to allocate points on both sides of the gap, consistently capturing the low-mass cluster. This iteration-wise notion of global coverage is a simple mechanism to reduce sampling bias in PH computations.

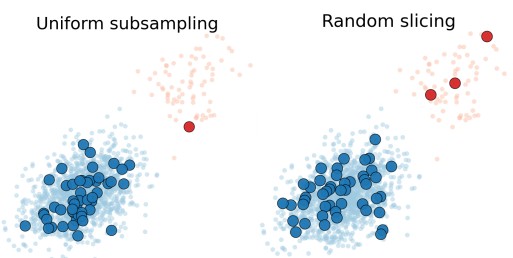

Figure 1: Uniform subsampling vs. random slicing on an imbalanced two-cluster cloud. Light points: full point set; highlighted points: a subsample ($s = 50$).

### 4.2 NW-Convolution Smoothing

Exact RKHS interpolation constructs the minimum-norm field matching anchors $v(x_i) = g_i$, but requires solving a kernel system at every step, which can dominate runtime when repeated in an optimization loop. We propose a fast alternative that preserves smoothness, yields explicit regularity control, and empirically induces update fields that are often better aligned with the geometric needs of persistence-driven optimization.

**Definition 4.2** (Gaussian weights and NW field). Fix a bandwidth $\sigma > 0$ and define the unnormalized Gaussian affinities

$$\phi_i(x) := \exp\left(-\frac{\|x - x_i\|_2^2}{2\sigma^2}\right), \qquad i \in I.$$

Let $Z(x) := \sum_{j \in I} \phi_j(x)$ and define normalized weights

$$w_i(x) := \frac{\phi_i(x)}{Z(x)}, \qquad \sum_{i \in I} w_i(x) = 1, \quad w_i(x) \in (0, 1). \tag{7}$$

The *Nadaraya-Watson (NW) convolution field* is

$$\bar{v}(x) := \sum_{i \in I} w_i(x)\, g_i \in \mathbb{R}^d. \tag{8}$$

First, we state the following characterization of the NW convolution field.

**Lemma 4.3** (Local kernel-regression characterization). *For each $x \in \mathbb{R}^d$, $\bar{v}(x)$ is the unique minimizer of the strictly convex problem*

$$\bar{v}(x) = \arg\min_{v \in \mathbb{R}^d} \sum_{i \in I} \phi_i(x)\, \|v - g_i\|_2^2. \tag{9}$$

Lemma 4.3 shows that NW is the *best local constant fit* to anchor gradients under Gaussian weights. This makes the role of $\sigma$ explicit: it controls the locality of the regression and hence the trade-off between (i) propagation of sparse information to non-anchors and (ii) fidelity to anchor updates. The proof is deferred to Section B.

Note that for every $x \in \mathbb{R}^d$, $\bar{v}(x)$ lies in the convex hull of $\{g_i : i \in I\}$, and by convexity of $\|\cdot\|_2$, $\|\bar{v}(x)\|_2 \leq \max_{i \in I} \|g_i\|_2$. Besides, the following lemma holds.

**Lemma 4.4** (Stability w.r.t. anchor gradients). *Fix anchor locations $\{x_i\}_{i \in I}$. Consider two anchor-gradient families $\{g_i\}$ and $\{g_i'\}$ and let $\bar{v}, \bar{v}'$ be the corresponding NW fields. Then*

$$\sup_{x \in \mathbb{R}^d} \|\bar{v}(x) - \bar{v}'(x)\|_2 \leq \max_{i \in I} \|g_i - g_i'\|_2.$$

Lemma 4.4 implies that NW smoothing is *non-expansive* in the anchor gradients: noise in sparse gradients is not amplified by the extension step. The proof is deferred to Section B.

In the following, for each anchor $i \in I$, define

$$\rho_i(\sigma) := \sum_{j \in I \setminus \{i\}} \exp\left(-\frac{\|x_i - x_j\|_2^2}{2\sigma^2}\right).$$

The following theorem studies when NW *nearly* interpolates, i.e., when the exact weight at anchors are close to the corresponding gradients.

**Theorem 4.5** (Anchor error bound). *Assume $\|g_i\|_2 \leq M$ for all $i \in I$. Then for each $i \in I$,*

$$\|\bar{v}(x_i) - g_i\|_2 \leq 2M \frac{\rho_i(\sigma)}{1 + \rho_i(\sigma)}. \tag{10}$$

Theorem 4.5 formalizes the *fidelity-propagation trade-off*: small $\sigma$ reduces cross-anchor mixing (small $\rho_i(\sigma)$) and yields near-interpolation, whereas larger $\sigma$ increases global propagation but mixes anchor directions.

In the following, we provide some regularity properties of $\bar{v}$. First, notice that $\bar{v}$ is $\mathcal{C}^\infty$ on $\mathbb{R}^d$. Indeed, each $\phi_i$ is $\mathcal{C}^\infty$ and $Z(x) > 0$ for all $x$ since all $\phi_i(x) > 0$. Thus $w_i = \frac{\phi_i}{Z}$ is $\mathcal{C}^\infty$ and $\bar{v} = \sum_i w_i g_i$ is $\mathcal{C}^\infty$. Furthermore, we have the following Lipschitz regularity property.

**Theorem 4.6** (Global Lipschitz constant of the NW field). *Assume $\|g_i\|_2 \leq M$ for all $i \in I$. Then $\bar{v}$ is globally Lipschitz and*

$$\|D\bar{v}(x)\|_{\mathrm{op}} \leq \frac{\mathrm{diam}(P)}{\sigma^2} M \qquad \forall x \in \mathbb{R}^d. \tag{11}$$

*In particular, $\|\bar{v}(x) - \bar{v}(y)\|_2 \leq L \|x - y\|_2$ for all $x, y$ with $L := \frac{\mathrm{diam}(P)M}{\sigma^2}$.*

The proof of Theorem 4.6 is deferred to Section B. The bound scales as $1/\sigma^2$: larger bandwidth yields smoother (more stable) deformations. The factor $\mathrm{diam}(P)$ captures the geometric spread of anchors: if anchors are confined to a small region, the induced field cannot vary too rapidly.

**Geometric intuitions on NW smoothing.**
To build intuition for the behavioral gap between exact kernel-based diffeomorphic interpolation and our NW-Convolution smoothing, we consider a deliberately simple anchor configuration (Fig. 2) that isolates the key mechanism: *normalization*. We place anchors in two tight clusters with opposite vectors, and compare the induced ambient fields for a fixed bandwidth $\sigma = 0.35$.

In the RKHS construction (used by Carrière et al. (2024)), the vector field takes the form $v_{\mathrm{ker}}(x) = \sum_j k_\sigma(x, x_j)\,\alpha_j$, where the coefficients $\alpha$ are chosen so that $v_{\mathrm{ker}}(x_i) = g_i$ at anchors. Because the expansion uses *unnor-*

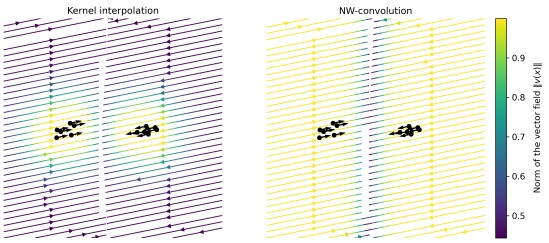

Figure 2: Kernel vs. NW smoothing on the same anchors. Two spatial clusters of anchors carry opposite directions. Left: exact Gaussian-RKHS interpolation (KERNEL). Right: NW smoothing (NW). Colors show the norm of the smoothed vector field.

*malized* Gaussians, the overall magnitude typically *decays away from the anchor set*: when $x$ is far from every $x_j$, all kernels $k_\sigma(x, x_j)$ become small simultaneously. This induces a field that is strongly localized around the regions that contain anchors (and is visually reflected by the attenuation of $\|v(x)\|$ away from the clusters).

In contrast, NW defines a convex combination of anchor vectors $v_{\mathrm{NW}}(x) = \sum_{i \in I} w_i(x) g_i$ (Definition 4.2), where the normalization enforces $\sum_j w_j(x) = 1$ for all $x$, so $v_{\mathrm{NW}}(x)$ behaves as a *local barycentric average* of the anchor gradients. Two qualitative consequences follow. First, the field does not vanish simply because $x$ is far from anchors: when all kernels are simultaneously small, the ratio defining $w_j(x)$ remains well-defined and tends to a near-uniform weighting across anchors, so the far-field approaches a (typically nonzero) average direction. Second, regions dominated by one cluster naturally inherit that cluster's direction, while the transition zone between clusters exhibits a smooth cancellation pattern (the averaging mediates the conflict rather than enforcing exact matching).

In topological optimization, topological gradients are sparse and highly local: only a few points (anchors) receive nonzero updates. Exact interpolation propagates these updates but can produce highly localized magnitude profiles and requires solving a kernel system. NW smoothing instead provides a cheap, stable propagation mechanism whose geometry is easy to interpret: it transports the anchor signal via normalized similarity weights, yielding a globally defined field with controlled mixing across competing anchors.

In the two-cluster setting of Figure 2, a natural desired smoothing is to propagate the influence of the opposing anchor groups across the domain so that points on each side are consistently driven in the direction prescribed by the nearest cluster, rather than receiving a near-zero update away from anchors. This effect is precisely what the normalized NW averaging promotes, and it helps explain its

empirical effectiveness (see Section 5) as a faster surrogate to exact kernel interpolation. Additional configurations illustrating the same phenomena are provided and discussed in Appendix C.

**Practical considerations: sensitivity to the bandwidth $\sigma$.** While NW-Convolution smoothing provides a fast and principled extension of sparse topological gradients, its behavior is strongly controlled by the bandwidth parameter $\sigma$, which sets the locality of averaging. To quantify this sensitivity, we run NW-smoothing with diffeomorphic updates on a noisy circle in $\mathbb{R}^2$ (Gaussian noise added to a unit circle) using the *collapser* loss, i.e. a persistence objective that penalizes large birth and death times and thus encourages rapid topological simplification (see Appendix D.2 for a formal definition of the loss and visualizations for this task). Figure 3 reports the loss trajectories for several fixed values of $\sigma$. Beyond the existence of a clearly favorable range (here around $\sigma = 0.1$), the curves exhibit substantial variability in both convergence speed

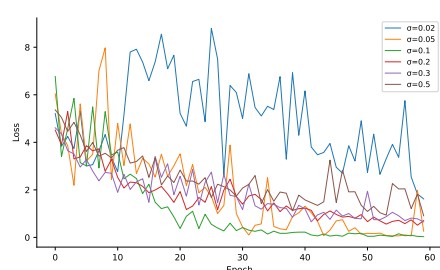

Figure 3: Loss trajectories (collapser loss) on a noisy circle in $\mathbb{R}^2$ for different fixed $\sigma$ values, using the same diffeomorphic update scheme.

and attained objective value, underscoring that $\sigma$ is a critical hyper-parameter rather than a minor adjustment. This observation motivates *learning* (or adaptively scheduling) $\sigma$ during optimization, we provide more details about this point in Section 5.

## 5 EXPERIMENTAL RESULTS

We evaluate our pipeline *subsampling via random slicing and NW-Convolution smoothing*[1] on a challenging 3D topological augmentation task. We consider the Stanford Bunny point cloud (Turk & Levoy, 1994), i.e. the mesh vertices, yielding an initial point set $X_0 = \{x_i\}_{i=1}^n \subset \mathbb{R}^3$, where $n = 35, 947$. We optimize a persistence-based objective that promotes prominent two-dimensional topological features, i.e. we encourage the formation (or strengthening) of a cavity by increasing its persistence.

**Loss.** We use the *bunny loss*, a topological augmentation objective defined by

$$\mathcal{L}_{\text{bunny}}(X) = - \sum_{\alpha \in \text{Dgm}_2(X)} \left( \tfrac{1}{2}(d_\alpha - b_\alpha) \right)^2 + \sum_{i=1}^n \max\left( \|x_i\|_\infty - 1, 0 \right), \tag{12}$$

i.e. a (negative) squared total persistence term in dimension 2, together with a soft box penalty that discourages points from leaving $[-1, 1]^3$.

**Learning the bandwidth $\sigma$.** The bandwidth $\sigma$ controls the locality of NW-Convolution smoothing and strongly impacts the resulting deformation (Figure 3). Rather than treating $\sigma$ as a fixed hyper-parameter, we can *learn* it online by backpropagating through the NW weights. Concretely, we parameterize a positive bandwidth as

$$\sigma(\theta) = \sigma_0 \exp(\theta), \tag{13}$$

with a scalar $\theta \in \mathbb{R}$. At each iteration, we first compute the raw sparse PH gradient $g_{\text{raw}}(X)$ (without internal smoothing), build the NW movement field $\bar{v}_\theta(X)$ using equation 13 inside the Gaussian weights, and form a one-step look-ahead point cloud

$$X^+(\theta) = X - \eta_X \, \bar{v}_\theta(X), \tag{14}$$

where $\eta_X$ is the point-update step size. We then update $\theta$ by minimizing the *look-ahead* loss

$$\theta \leftarrow \theta - \eta_\sigma \, \nabla_\theta \, \mathcal{L}\big(X^+(\theta)\big), \tag{15}$$

---

[1]In our plots we label our approach as NW-CONVOLUTION, this always refers to the *combination* of NW smoothing *and* random slicing subsampling.

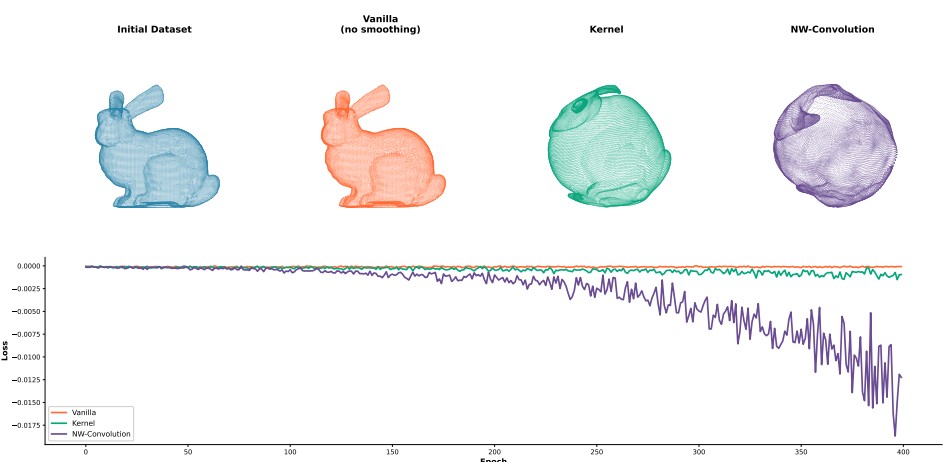

Figure 4: 3D bunny augmentation task. Top: initial point cloud (Stanford Bunny) and point clouds after 400 epochs for vanilla sparse updates (VANILLA), exact kernel-based diffeomorphic interpolation (KERNEL), and our approach (NW-CONVOLUTION, i.e., random slicing subsampling and NW smoothing). Bottom: loss curves versus epochs for the three methods.

Table 1: **Bunny augmentation: best loss and runtime.** Best objective value (lower is better) and average time per epoch, averaged over the full run.

| Method | Best Loss $\downarrow$ | Avg Time/Epoch (s) $\downarrow$ |
|---|---|---|
| Vanilla | -0.00026 | **2.044** |
| Kernel | -0.00151 | 19.979 |
| **NW-Convolution (ours)** | **-0.01869** | 2.573 |

with $\eta_\sigma$ a separate learning rate. In implementation, we stop gradients through $X$ and $g_{\text{raw}}$ when differentiating equation 15 (i.e., treat them as constants during the $\theta$ update), which yields a stable and inexpensive meta-gradient while still adapting $\sigma$ to the local optimization regime. This procedure can be interpreted as selecting the bandwidth that makes the *next* deformation step most beneficial for the topological objective, and we use it as a practical alternative to manual tuning when the optimal $\sigma$ varies substantially across tasks and datasets. Pseudo-codes of the algorithms for learning $\sigma$ and our NW-Convolution are given in Section A.

**Compared methods.** We compare: (i) **Vanilla** topological gradient descent, which updates only anchor points (sparse gradient); (ii) **Kernel** diffeomorphic interpolation Carrière et al. (2024), using exact Gaussian-RKHS interpolation; and (iii) **NW-Convolution (ours)**, which replaces the kernel solve with NW smoothing and uses random slicing for subsampling.

**Hyper-parameters.** For all methods we run $T = 400$ optimization steps with subsampling size $s = 100$ and compute persistent homology in dimension 2. Vanilla and kernel-based diffeomorphic interpolation use SGD with learning rate $\eta_X = 0.1$, for the kernel baseline we set the Gaussian bandwidth to $\sigma = 0.05$. For our approach, we use the same subsampling budget but replace uniform subsampling by projection-stratified sampling and apply the NW-Convolution smoother with base bandwidth $\sigma_0 = 5 \times 10^{-3}$ and step size $\eta_X = 0.18$.

**Results.** Figure 4 shows the initial bunny and the final point clouds after 400 epochs for the three optimizers, together with the loss evolution. Table 1 reports the best loss achieved over the run and the average runtime per epoch. Our method achieves substantially lower objective values than both baselines while remaining close to the runtime of the vanilla sparse update. In particular, NW smoothing introduces only a small overhead compared to no smoothing, whereas exact kernel interpolation is an order of magnitude slower in this experiment.

Further experiments in 2D can be found in Appendix D.

## 6 CONCLUSION

We revisited a central practical bottleneck in persistence-based topological optimization: the sparsity and instability of topological gradients when PH is computed on subsamples. Building on the ambient-smoothing viewpoint of diffeomorphic updates, we proposed a lightweight alternative to exact kernel interpolation: a Nadaraya-Watson Gaussian convolution that extends sparse anchor gradients without solving a kernel system, together with a simple projection-stratified (random slicing) subsampling scheme to improve geometric coverage. Across our benchmarks in 2D and 3D, this combination yields substantially faster iterations than kernel-based diffeomorphic interpolation while achieving improved objective values. A key limitation of the present work is that we focus on controlled synthetic and standard geometry benchmarks, we do not yet demonstrate end-to-end impact on downstream real-world tasks. We view our contribution primarily as a step towards making persistence-driven optimization more scalable and, ultimately, more practical in application settings where large point sets and repeated PH computations are unavoidable.

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

# A  ALGORITHMS

## A.1  PROJECTION-STRATIFIED SUBSAMPLING (RANDOM SLICING)

---

**Algorithm 1** Projection-stratified subsampling (random slicing)

---

**Require:** Point cloud $X = \{x_i\}_{i=1}^n \subset \mathbb{R}^d$, subsample size $s$
1: Sample $u \sim \mathcal{N}(0, I_d)$ and normalize $u \leftarrow u/\|u\|_2$
2: Compute projections $t_i \leftarrow \langle x_i, u \rangle$ for $i = 1, \ldots, n$
3: Let $\pi$ be the permutation sorting $(t_i)$ increasingly
4: **for** $k = 0$ **to** $s - 1$ **do**
5:   $r_k \leftarrow \lfloor k(n-1)/(s-1) \rfloor$   /* evenly spaced ranks */
6:   Add index $\pi(r_k)$ to $S$
7: **end for**
8: **return** $S$

---

## A.2  LEARNING THE BANDWIDTH $\sigma$ (ONE-STEP LOOK-AHEAD)

---

**Algorithm 2** LEARNSIGMASTEP: one-step look-ahead update for $\sigma$

---

**Require:** Current cloud $X \in \mathbb{R}^{n \times d}$, sparse gradient $g \in \mathbb{R}^{n \times d}$, anchors $I$
**Require:** Base bandwidth $\sigma_0$, current $\theta$   /* so that $\sigma = \sigma_0 \exp(\theta)$ */
**Require:** Step sizes $\eta_X$ (for $X$) and $\eta_\sigma$ (for $\theta$), subsample $S$
1: $\sigma \leftarrow \sigma_0 \exp(\theta)$   /* enforce positivity */
2: $X^{\text{sg}} \leftarrow \text{stopgrad}(X)$, $g^{\text{sg}} \leftarrow \text{stopgrad}(g)$
3: Build $\bar{v}_\theta(\cdot)$ from $(x_i, g_i^{\text{sg}})_{i \in I}$ using NW weights with bandwidth $\sigma$
4: $\tilde{X} \leftarrow X^{\text{sg}} - \eta_X \bar{v}_\theta(X^{\text{sg}})$   /* look-ahead point cloud */
5: $\theta \leftarrow \theta - \eta_\sigma \nabla_\theta \ell(\text{Dgm}(\tilde{X}_S))$   /* meta-gradient step */
6: **return** $\sigma_0 \exp(\theta)$

---

## A.3  NW-CONVOLUTION UPDATE WITH RANDOM SLICING (OURS)

---

**Algorithm 3** NW-Convolution smoothing with random slicing (ours)

---

**Require:** Initial $X_0 \in \mathbb{R}^{n \times d}$, loss $\ell$, bandwidth $\sigma$ /* fixed, or learned via Alg. 2 */
**Require:** Step size $\eta_X$, subsample size $s$, epochs $T$
1: **for** $t = 0$ **to** $T - 1$ **do**
2:   $S_t \leftarrow \text{RANDOMSLICING}(X_t, s)$ /* Alg. 1 */
3:   Compute PH on $X_t[S_t]$ and sparse gradient $g^{(t)} = \nabla \ell(\text{Dgm}(X_t[S_t]))$
4:   $I_t \leftarrow \{i \in S_t : g_i^{(t)} \neq 0\}$   /* anchor indices */
5:   **if** learning $\sigma$ **then**
6:     $\sigma \leftarrow \text{LEARNSIGMASTEP}(X_t, g^{(t)}, I_t, \sigma)$   /* Alg. 2 */
7:   **end if**
8:   **for all** $x \in X_t$ **do**   /* evaluate NW field on all points */
9:     $\phi_i \leftarrow \exp\big(-\|x - x_i\|_2^2/(2\sigma^2)\big)$ for all $i \in I_t$
10:     $Z \leftarrow \sum_{j \in I_t} \phi_j$
11:     $\bar{v}_t(x) \leftarrow \sum_{i \in I_t} (\phi_i/Z) g_i^{(t)}$
12:     $x \leftarrow x - \eta_X \bar{v}_t(x)$
13:   **end for**
14: **end for**
15: **return** $X_T$

---

## B    OMITTED PROOFS

*Proof of Lemma 4.1.* Let $\Delta = \frac{n-1}{s-1}$. Since $r_k = \lfloor k\Delta \rfloor$,

$$r_{k+1} \le (k+1)\Delta,$$

and

$$-r_k < -(k\Delta - 1).$$

Hence

$$r_{k+1} - r_k =< \Delta + 1,$$

hence

$$r_{k+1} - r_k \le \lceil \Delta \rceil,$$

which yields the bound. $\qquad\square$

*Proof of Lemma 4.3.* Let

$$F_x(v) := \sum_{i \in I} \phi_i(x) \|v - g_i\|_2^2.$$

Expanding,

$$F_x(v) = \left( \sum_i \phi_i(x) \right) \|v\|_2^2 - 2\langle v, \sum_i \phi_i(x)g_i \rangle + C(x),$$

hence $F_x$ is strictly convex and differentiable with

$$\nabla_v F_x(v) = 2Z(x)v - 2\sum_i \phi_i(x)g_i.$$

Setting $\nabla_v F_x(v) = 0$ yields

$$v = \sum_i \left( \frac{\phi_i(x)}{Z(x)} \right) g_i = \sum_i w_i(x)g_i = \bar{v}(x).$$

$\square$

*Proof of Lemma 4.4.* For any $x$,

$$\bar{v}(x) - \bar{v}'(x) = \sum_i w_i(x)(g_i - g_i').$$

Taking norms and using $\sum_i w_i(x) = 1$,

$$\|\bar{v}(x) - \bar{v}'(x)\|_2 \le \sum_i w_i(x)\|g_i - g_i'\|_2$$

$$\le \max_i \|g_i - g_i'\|_2.$$

$\square$

*Proof of Theorem 4.5.* For each $i \in I$, we have,

$$w_i(x_i) = \frac{1}{1 + \rho_i(\sigma)},$$

and,

$$\sum_{j \ne i} w_j(x_i) = \frac{\rho_i(\sigma)}{1 + \rho_i(\sigma)}$$

Write $\bar{v}(x_i) = w_i(x_i)g_i + \sum_{j \ne i} w_j(x_i)g_j$. Then

$$\bar{v}(x_i) - g_i = (w_i(x_i) - 1)g_i + \sum_{j \ne i} w_j(x_i)g_j.$$

Taking norms and using $\|g_j\|_2 \leq M$,

$$\|\bar{v}(x_i) - g_i\|_2 \leq (1 - w_i(x_i)) \|g_i\|_2 + \sum_{j \neq i} w_j(x_i) \|g_j\|_2$$

$$\leq M(1 - w_i(x_i)) + M \sum_{j \neq i} w_j(x_i).$$

Furthermore,

$$1 - w_i(x_i) = \frac{\rho_i(\sigma)}{1 + \rho_i(\sigma)} \text{ and } \sum_{j \neq i} w_j(x_i) = \frac{\rho_i(\sigma)}{1 + \rho_i(\sigma)},$$

yielding equation 10.                                                                               $\square$

*Proof of Theorem 4.6.* We start by stating and proving the following lemma which provides an expression of the gradient of the weights and bounds its norm .

**Lemma B.1** (Gradient of weights). *Let $\bar{x}(x) := \sum_{j \in I} w_j(x) x_j$ be the weighted barycenter of anchors at $x$. Then for all $i \in I$ and $x \in \mathbb{R}^d$,*

$$\nabla w_i(x) = \frac{w_i(x)}{\sigma^2} (x_i - \bar{x}(x)). \tag{16}$$

*Moreover, for all $x$ and $i$,*

$$\|\nabla w_i(x)\|_2 \leq \frac{\text{diam}(P)}{\sigma^2} w_i(x),$$

*where* $\text{diam}(P) := \max_{p,q \in I} \|x_p - x_q\|_2$

*Proof of Lemma B.1.* Using

$$\phi_i(x) = \exp\left(-\frac{\|x - x_i\|^2}{2\sigma^2}\right) \quad \text{and} \quad w_i = \frac{\phi_i}{Z},$$

we have,

$$\nabla \phi_i(x) = -(x - x_i)\frac{\phi_i(x)}{\sigma^2},$$

and

$$\nabla Z(x) = \sum_j \nabla \phi_j(x) = -\frac{1}{\sigma^2} \sum_j (x - x_j)\phi_j(x).$$

By the quotient rule,

$$\nabla w_i(x) = \frac{\nabla \phi_i(x) Z(x) - \phi_i(x) \nabla Z(x)}{Z(x)^2}$$

$$= -\frac{\phi_i(x)}{\sigma^2 Z(x)}\left[(x - x_i) - \sum_j w_j(x)(x - x_j)\right].$$

Since

$$\sum_j w_j(x)(x - x_j) = x - \sum_j w_j(x) x_j$$

$$= x - \bar{x}(x),$$

we obtain equation 16.

Hence,

$$\|\nabla w_i(x)\|_2 = \frac{w_i(x)}{\sigma^2}\|x_i - \bar{x}(x)\|_2.$$

Since $\bar{x}(x)$ is a convex combination of points in $P$, it lies in $\text{Conv}(P)$, hence

$$\|x_i - \bar{x}(x)\|_2 \leq \text{diam}(P).$$

$\square$

Differentiate $\bar{v}(x) = \sum_i w_i(x) g_i$:

$$D\bar{v}(x) = \sum_{i \in I} (\nabla w_i(x)) \otimes g_i.$$

Using $\|a \otimes b\|_{\mathrm{op}} = \|a\|_2 \|b\|_2$,

$$\|D\bar{v}(x)\|_{\mathrm{op}} \leq \sum_i \|\nabla w_i(x)\|_2 \|g_i\|_2$$

$$\leq M \sum_i \|\nabla w_i(x)\|_2.$$

Apply Lemma B.1:

$$\sum_i \|\nabla w_i(x)\|_2 \leq \frac{\mathrm{diam}(P)}{\sigma^2} \sum_i w_i(x)$$

$$= \frac{\mathrm{diam}(P)}{\sigma^2},$$

giving equation 11. The Lipschitz inequality follows from the mean value theorem. $\square$

## C  ADDITIONAL KERNEL VS. NW VISUALIZATIONS

Figure 5: **Kernel interpolation vs. NW smoothing under four anchor configurations.** In each panel, the left sub-plot is exact Gaussian-RKHS interpolation (KERNEL) and the right sub-plot is Nadaraya-Watson smoothing (NW). Colors represent $\|v(x)\|$. All cases use the same anchor positions, anchor vectors, and bandwidth $\sigma$ within a panel.

Fig. 5 highlights systematic geometric differences between exact kernel interpolation and NW smoothing. The most robust distinction is that KERNEL produces an *unnormalized* Gaussian expansion, so $\|v(x)\|$ tends to *decay away from anchors*, whereas NW computes a *normalized* convex combination of anchor vectors, yielding a field that behaves like a *local barycentric average* and therefore need not vanish in the far-field.

**Ring conflict.** This configuration resembles the qualitative behavior of persistence-driven updates: multiple anchors distributed on a ring induce a swirl-like motion, while a few interior anchors introduce competing directions. KERNEL tends to concentrate magnitude near the anchor set (visible attenuation toward the boundary of the window), and the flow lines exhibit sharper steering to satisfy exact anchor constraints. NW yields a more uniformly energized outer region and a smoother transition between the ring-induced swirl and the interior conflict, reflecting local averaging rather than exact matching.

**Dipole.** A small number of strong, antagonistic anchors produce a dipole-like geometry. KERNEL emphasizes the dominance of strong anchors by producing localized high-norm lobes near them, with rapid attenuation elsewhere. NW spreads influence more evenly: large regions inherit the prevailing direction under normalized weighting, resulting in wider basins of attraction/repulsion and smoother separatrices.

**Saddle ring.** Here, a ring component combines with a central saddle structure. KERNEL produces several localized vortical/saddle patterns whose intensities decay away from anchors. NW preserves the qualitative topology of the flow but exhibits a more "global" strength profile and smoother interactions between the saddle and ring components, consistent with a convex combination mechanism.

**Random sparse.** With irregular anchor placement, KERNEL can generate pronounced local features around isolated anchors, with relatively weak far-field magnitude. NW is notably robust:

because each evaluation is a normalized average over nearby anchors, the field remains well-defined and smooth across the domain, often producing globally coherent motion even when anchors are unevenly distributed.

Overall, these visualizations support two practical messages relevant to persistence-based optimization: (i) NW smoothing behaves like a geometry-aware averaging operator that propagates sparse topological gradients beyond anchors with minimal computational overhead, and (ii) normalization changes the far-field behavior compared to RKHS interpolation, which can be beneficial for spreading topological updates across the ambient point set.

# D    FURTHER EXPERIMENTS IN 2D

We report two additional 2D experiments that complement the main text and further illustrate the empirical behavior of our scalable pipeline (NW-CONVOLUTION, i.e. projection-stratified subsampling with NW smoothing) against (i) vanilla topological gradient descent (sparse updates) and (ii) kernel-based diffeomorphic interpolation. For both experiments, we use $N = 1000$ points, subsample size $s = 100$, and optimize with step size $\eta_X = 0.05$. For our method we use a base bandwidth $\sigma_0 = 0.1$ and, when enabled, a one-step look-ahead update of $\sigma$ with learning rate $\eta_\sigma = 10^{-3}$ (cf. Section A.2), while for diffeomorphic interpolation, we use a bandwidth $\sigma = 0.1$.

## D.1    MAXPERS ON A UNIFORM SQUARE

We first consider a point cloud initialized uniformly on $[-1, 1]^2$ and optimize a standard topological *augmentation* objective that encourages the emergence of persistent $H_1$ features while keeping points within the box. Concretely, given $\mathrm{Dgm}_1(X) = \{(b_\alpha, d_\alpha)\}$, we use

$$\mathcal{L}_{\mathrm{maxpers}}(X) = - \sum_{\alpha \in \mathrm{Dgm}_1(X)} \left( \tfrac{1}{2}(d_\alpha - b_\alpha) \right)^2 + \sum_{i=1}^{N} \max\big(\|x_i\|_\infty - 1,\, 0\big), \qquad (17)$$

i.e. a negative squared total persistence term in dimension 1 (encouraging long-lived cycles) with a soft constraint that discourages leaving $[-1, 1]^2$. We optimize for $T = 300$ epochs.

Figure 6 shows the initial and final configurations together with the loss curves. The vanilla method remains limited by the sparsity of the topological gradient, whereas both smoothing-based methods propagate the sparse signal to non-anchor points and more reliably shape the cloud toward configurations with stronger $H_1$ persistence. In this setting, NW-CONVOLUTION achieves the lowest objective value while being markedly faster per epoch than kernel-based interpolation (Table 2).

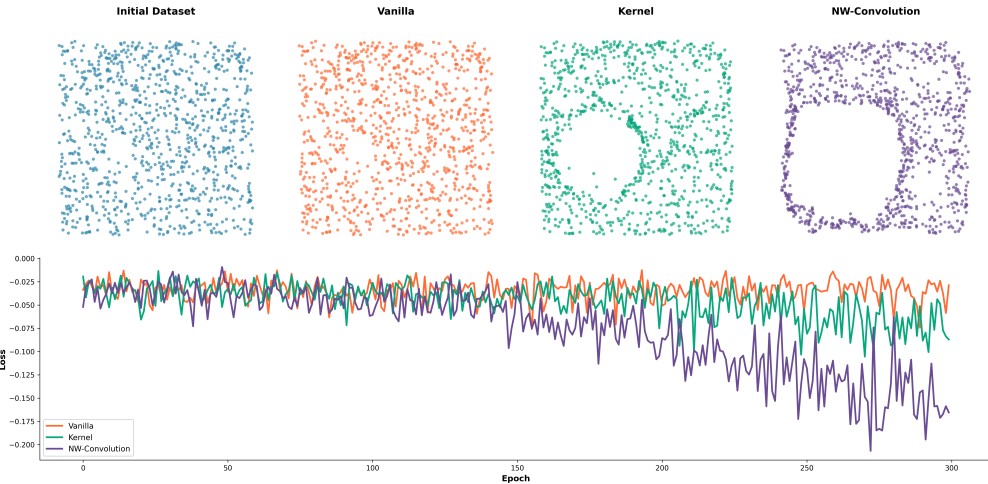

Figure 6: **2D maxpers from a uniform square.** Top: initial cloud sampled uniformly in $[-1, 1]^2$ and final configurations after $T = 300$ epochs for vanilla updates, kernel-based diffeomorphic interpolation (KERNEL), and our method (NW-CONVOLUTION = random slicing + NW smoothing). Bottom: loss trajectories for equation 17 (lower is better).

## D.2    COLLAPSER ON A NOISY CIRCLE

We next consider a complementary *simplification* task: the initial cloud is a noisy circle in $\mathbb{R}^2$, and we optimize the `collapser` loss, which penalizes both births and deaths and therefore pushes topological features toward short scales. In our implementation this reads

$$\mathcal{L}_{\mathrm{coll}}(X) = \sum_{\alpha \in \mathrm{Dgm}_1(X)} (d_\alpha + b_\alpha)^2, \qquad (18)$$

Table 2: **2D maxpers: best loss and runtime.** Best objective value (lower is better) and average time per epoch.

| Method | Best Loss ↓ | Avg Time/Epoch (s) ↓ |
|---|---|---|
| Vanilla | -0.0758 | 0.5615 |
| Kernel | -0.1053 | 0.6895 |
| **NW-Convolution (ours)** | **-0.2066** | **0.3475** |

so that driving $\mathcal{L}_{\text{coll}}$ down encourages both early death and early birth, i.e. rapid topological collapse.

Figure 7 shows that NW-CONVOLUTION leads to a substantially faster and more decisive decrease of equation 18, and qualitatively produces a sharper collapse of the noisy ring than both baselines. Quantitatively, Table 3 confirms that our method achieves the lowest loss while remaining close to the runtime of the vanilla sparse updates, and significantly faster than kernel-based interpolation.

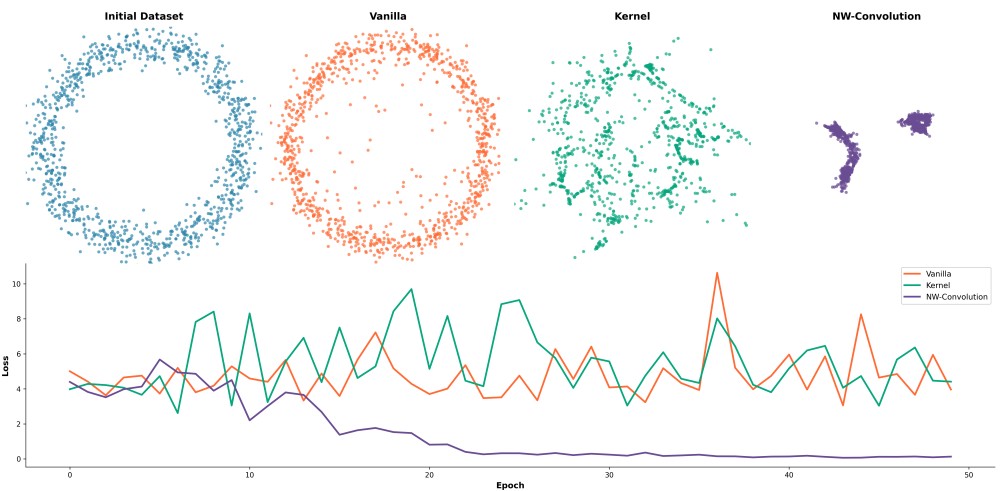

Figure 7: **2D collapser from a noisy circle.** Top: initial noisy circle and final point clouds after $T = 50$ optimization epochs using vanilla updates, kernel-based diffeomorphic interpolation (KERNEL), and our method (NW-CONVOLUTION = random slicing + NW smoothing). Bottom: loss curves for the collapser objective equation 18 (lower is better).

Table 3: **2D collapser: best loss and runtime.** Best objective value (lower is better) and average time per epoch.

| Method | Best Loss ↓ | Avg Time/Epoch (s) ↓ |
|---|---|---|
| Vanilla | 3.057 | **0.5233** |
| Kernel | 2.615 | 0.8800 |
| **NW-Convolution (ours)** | **0.064** | 0.5406 |

