# OpenReview forum: "Towards Scalable Persistence-Based Topological Optimization"
_ICLR.cc/2026/Workshop/GRaM — ICLR 2026 Workshop GRaM Poster_

### Official Review · Reviewer_L8LC · 2026-02-09
**The authors propose a scalable approach to persistence-based topological optimization that is simple but effective.**

**Rating:** 4
**Confidence:** 4

**Review:**

The authors propose two optimizations to the persistence-based topological optimization pipeline. Both are simple adjustments to existing ideas, which limits the level of novelty; however, they are well-motivated and executed well. There is accompanying theory that is nice to support these ideas, but it isn't critical to the claim.

Pros:
- Simple, practical modifications that are effective and well executed.
- NW smoothing is computationally efficient and has a nice interpretation.
- The tuning of the Gaussian parameter $\sigma$ in this smoothing is done efficiently but logically.

Cons:
- Limited conceptual novelty. Both contributions feel like natural engineering adjustments to existing pipelines rather than fundamentally new ideas. Both smoothing and subsampling already exist in topological optimization, though these are more practical.
- It is unclear why random-direction slicing with evenly spaced ranks is preferable to other plausible strategies. For instance, using data-adaptive axes, clustering and then sampling within each cluster, or using Chebyshev node-style sampling (because the endpoints of each shape are likely quite critical).

Note
- Figure 4 on page 8 is much too high quality and makes reading the manuscript difficult on my computer (due to freezing).

In optimization 1, the authors replace RKHS-based kernel interpolation with Nadaraya--Watson (NW) Gaussian convolution to smooth sparse anchor gradients into a global vector field. This serves the same purpose as prior approaches (i.e., the smoothing idea is not new), but is computationally cheaper. In optimization 2, instead of uniform subsampling, they select equally spaced points along a random direction.

In my view, the most novel aspect of the paper is the adaptive tuning of the Gaussian parameter $\sigma$. The authors parameterize it as $\sigma(\theta) = \sigma_0 e^\theta$,
which ensures $\sigma > 0$, and update $\theta$ online using a one-step look-ahead objective. To stay lightweight, they treat the current point cloud and sparse gradients as fixed, construct the NW-smoothed update field using the current $\sigma$, simulate a single optimization step forward, and then adjust $\theta$ by differentiating the loss evaluated at this hypothetical next state.

While the paper is well-executed and the optimizations are sound and useful, the novelty is low, and I would like to be convinced that these optimizations are the best possible versions (see con 2). As this is a paper on improvements to existing work, it would be much stronger if it could comment on whether other heuristics are better or comparable.

**Pmlr Suitability:**

Yes

---

### Official Review · Reviewer_7eMS · 2026-02-11

**Rating:** 5
**Confidence:** 3

**Review:**

The paper introduces several modifications to improve persistence-based topological optimization: (1) random slicing subsampling, (2) Nadaraya-Watson (NW) gradient smoothing, and (3) learning the smoothing bandwidth parameter $\sigma$. I think the paper aligns well with the theme of the workshop.

That said, the experimental section is limited in scope. As all components are presented as improvements over existing approaches, the strength of the contributions depends a lot on the empirical validation in my opinion. For example, gradient smoothing itself is not new, but the novelty is the NW-based smoothing. Subsampling is not new, but the proposed slicing strategy is. While the design choices are well explained and well motivated, they are not evaluated in a sufficiently exhaustive way.

The following issues should be addressed in my opinion:
1. All three components (random slicing, NW smoothing, bandwidth learning) are only evaluated in a single combined experiment. It would be important to study the individual contribution of these modifications. For example, compare kernel-based smoothing and NW convolution under standard random subsampling without bandwidth learning.
2. I think that bandwidth learning should be applicable to kernel-based methods as well? Including this variant would yield a fairer comparison when evaluating against the proposed method.
3. Lemma 4.1 ("Maximal Rank Gap") effectively is just the definition of "evenly spaced". Presenting it as a mathematical result should be reconsidered.
4. Limitations or potential failure modes of NW gradient smoothing are not discussed.
5. Figure 1: Please also show the random direction and not only the sampled points

Also, the literature on kernel-based methods (fast kernel summation) includes a range of other slicing strategies. In particular, quasi-Monte-Carlo (QMC) methods are frequently used (see, e.g., [1–3]). Incorporating QMC-based designs into the slicing procedure could maybe further improve the approach and could be worth considering.

References:
- [1] Avron et al. Quasi-Monte Carlo Freature Maps for Shift-Invariant Kernels (2015)
- [2] Hertrich et al. Fast Summation of Radial Kernels via QMC Slicing (2025), ICLR
- [3] Brauchart et al. QMC designs: optimal order quasi Monte Carlo integration schemes on the sphere (2024)

**Pmlr Suitability:**

Yes

---

### Official Review · Reviewer_5KHT · 2026-02-23
**The paper introduces an efficient method for persistence-based optimization, but requires ablation studies separating NW smoothing from random slicing and further validation to fully justify its empirical claims.**

**Rating:** 6
**Confidence:** 3

**Review:**

### Summary
The paper addresses the computational and geometric bottlenecks of persistence-based topological optimization, specifically gradient sparsity and subsampling bias. It proposes a scalable pipeline that replaces expensive exact RKHS diffeomorphic interpolation with a fast, normalized Nadaraya-Watson (NW) Gaussian convolution. Additionally, it introduces "random slicing," a lightweight 1D projection-based subsampling technique to improve geometric coverage and mitigate the density bias of uniform sampling. The method is evaluated on 2D and 3D point cloud optimization tasks.

### Strengths
* **Well-Motivated and Clearly Written:** The paper does an excellent job of isolating the specific flaws of exact kernel methods (computational cost and far-field vanishing gradients) and uniform sampling (density bias).
* **Strong Empirical Results:** The proposed pipeline achieves significantly better objective values while running much faster than exact kernel interpolation on tasks like the 3D Stanford Bunny augmentation and 2D collapser.
* **Rigorous Theoretical Grounding:** The introduction of the NW convolution is backed by solid theoretical guarantees. The authors provide bounds for anchor approximation (Theorem 4.5), global Lipschitz continuity (Theorem 4.6), and stability with respect to anchor gradients (Lemma 4.4).

### Weaknesses
* **Confounded Contributions (Missing Ablation Study):** The core empirical weakness is that the proposed method, labeled "NW-CONVOLUTION" in the experiments, strictly bundles *both* random slicing and NW smoothing together. There is no ablation study to separate these contributions. It is currently impossible to tell whether the improved optimization trajectories are primarily due to the better geometric coverage of random slicing or the normalized propagation of the NW smoother. The authors should include cross-ablations (e.g., Uniform Sampling + NW Smoothing, and Random Slicing + Exact Kernel).
* **Limited to Toy/Synthetic Benchmarks:** The empirical evaluation relies entirely on standard geometric point clouds. While these are good proofs of concept, the authors admit in the conclusion that they do not demonstrate the method's impact on downstream, real-world machine learning tasks.
* **Hyperparameter Sensitivity and Overhead:** As shown in Figure 3, the NW smoothing is highly sensitive to the bandwidth parameter $\sigma$, which can drastically alter convergence. While the authors propose a "one-step look-ahead" meta-learning update to adapt $\sigma$ dynamically, this introduces its own hyperparameters (like the meta-learning rate $\eta_\sigma$) and computational overhead. The paper lacks a detailed discussion on how fragile this meta-update is compared to a fixed $\sigma$.

**Pmlr Suitability:**

Yes

---

### Meta-Review · Area_Chair_wQhu · 2026-02-25

**Decision:**

Accept

**Metareview:**

While two reviewers leaned towards rejection due to a lack of ablation studies and incremental conceptual novelty, I would like to overrule them to accept the paper. The method provides a practical, computationally efficient pipeline for persistence-based optimization backed by theoretical guarantees. For our GRaM workshop setting, I consider this solid engineering and theoretical contribution outweighs the need for exhaustive component ablations.

**Relevance To Proceedings:**

Yes — suitable for PMLR (long paper)

**Relevance To Workshop:**

Yes — suitable for GRaM

---

### Decision · Program_Chairs · 2026-03-02

Accept (Poster)